# Morphology, Development and Deformation of the Spine in Mild and Moderate Scoliosis: Are Changes in the Spine Primary or Secondary?

**DOI:** 10.3390/jcm10245901

**Published:** 2021-12-16

**Authors:** Theodoros B. Grivas, George Vynichakis, Michail Chandrinos, Christina Mazioti, Despina Papagianni, Aristea Mamzeri, Constantinos Mihas

**Affiliations:** 1Department of Orthopedics & Traumatology, “Tzaneio” General Hospital of Piraeus, 185 36 Piraeus, Greece; vini_gio@windowslive.com (G.V.); ChandrinosMichail@gmail.com (M.C.); 2Health Visitor, “Tzaneio” General Hospital of Piraeus, 185 36 Piraeus, Greece; maziotix@gmail.com; 3School Nurse—Health Visitor, Special Primary School of Rafina, 190 09 Attica, Greece; papdes2009@hotmail.com; 4Health Visitor, TOMY Attica Square, 104 45 Athens, Greece; mamzeri_aristea@hotmail.com; 5Department of Internal Medicine, Kymi General Hospital—Health Centre, 340 03 Euboea (Evia), Greece; gas521@yahoo.co.uk

**Keywords:** idiopathic scoliosis, scoliometer, truncal asymmetry, lateral spinal profile, surface topography, aetiology

## Abstract

Introduction and aim of the study: We aim to determine whether the changes in the spine in scoliogenesis of idiopathic scoliosis (IS), are primary/inherent or secondary. There is limited information on this issue in the literature. We studied the sagittal profile of the spine in IS using surface topography. Material and methods: After approval of the ethics committee of the hospital, we studied 45 children, 4 boys and 41 girls, with an average age of 12.5 years (range 7.5–16.4 years), referred to the scoliosis clinic by our school screening program. These children were divided in two groups: A and B. Group A included 17 children with IS, 15 girls and 2 boys. All of them had a trunk asymmetry, measured with a scoliometer, greater than or equal to 5 degrees. Group B, (control group) included 26 children, 15 girls and 11 boys, with no trunk asymmetry and scoliometer measurement less than 2 degrees. The height and weight of children were measured. The Prujis scoliometer was used in standing Adam test in the thoracic (T), thoraco-lumbar (TL) and lumbar (L) regions. All IS children had an ATR greater than or equal to 5 degrees. The Cobb angle was assessed in the postero-anterior radiographs in Group A. A posterior truncal surface topogram, using the “Formetric 4” apparatus, was also performed and the distance from the vertebra prominence (VP) to the apex of the kyphosis (KA), and similarly to the apex of the lumbar lordosis (LA) was calculated. The ratio of the distances (VP-KA) for (PV-LA) was calculated. The averages of the parameters were studied, and the correlation of the ratio of distances (VP-KA) to (VP-KA) with the scoliometer and Cobb angle measurements were assessed, respectively (Pearson corr. Coeff. r), in both groups and between them. Results: Regarding group A (IS), the average height was 1.55 m (range 1.37, 1.71), weight 47.76 kg (range 33, 65). The IS children had right (Rt) T or TL curves. The mean T Cobb angle was 24 degrees and 26 in L. In the same group, the kyphotic apex (KA (VPDM)) distance was −125.82 mm (range −26, −184) and the lordotic apex (LA (VPDM)) distance was −321.65 mm (range −237, −417). The correlations of the ratio of distances (KA (VPDM))/(LA (VPDM)) with the Major Curve Cobb angle measurement and scoliometer findings were non-statistically significant (Pearson r = 0.077, −0.211, *p*: 0.768, 0.416, respectively. Similarly, in the control group, KA (VPDM))/(LA (VPDM) was not significantly correlated with scoliometer findings (Pearson r = −0.016, −*p*: 0.939). Discussion and conclusions: The lateral profile of the spine was commonly considered to be a primary aetiological factor of IS due to the fact that the kyphotic thoracic apex in IS is located in a higher thoracic vertebra (more vertebrae are posteriorly inclined), thus creating conditions of greater rotational instability and therefore greater vulnerability for IS development. Our findings do not confirm this hypothesis, since the correlation of the (VP-KA) to (VP-KA) ratio with the truncal asymmetry, assessed with the scoliometer and Cobb angle measurements, is non-statistically significant, in both groups A and B. In addition, the aforementioned ratio did not differ significantly between the two groups in our sample (0.39 ± 0.11 vs. 0.44 ± 0.08, *p*: 0.134). It is clear that hypokyphosis is not a primary causal factor for the commencing, mild or moderate scoliotic curve, as published elsewhere. We consider that the small thoracic hypokyphosis in developing scoliosis adds to the view that the reduced kyphosis, facilitating the axial rotation, could be considered as a permissive factor rather than a causal one, in the pathogenesis of IS. This view is consistent with previously published views and it is obviously the result of gravity, growth and muscle tone.

## 1. Introduction

Idiopathic scoliosis (IS) is defined as a three-dimensional (3D) structural deformity of the spine and is associated with asymmetries of the trunk and the extremities [1]. The aetiology of this deformity is not yet clear, because there is no established theory of scoliogenesis. Adolescent idiopathic scoliosis (AIS) is the most common type of IS, and begins in early puberty. The incidence of AIS is 1–4% in adolescents and affects mainly females [2]. The frontal plane spinal curvature in IS is commonly assessed in the postero-anterior standing radiographs using the Cobb angle [3]. Similarly, the thoracic kyphosis is commonly assessed in the lateral standing radiographs using the Cobb method, usually from T4 to T12 vertebrae.

A major question in the study of IS aetiology is whether the growth changes in the spine in initiating and mild cases, are primary/inherent or secondary. There is no clear answer and there is limited information on this issue in the peer review literature. Some maintain that pathology begins within the spine [4]. while others argue that changes in the spine are secondary [5]. The research approach to shed some light in this issue is multidimensional, with the study of the various anatomical components of the deformity, such as the thoracic cage and the lateral vertebral profile.

As concerns the thoracic cage in IS, and especially in adolescents, there is the view that the rib length asymmetry in apical region is a secondary event to the scoliosis deformity and not a protagonist in the aetiopathogenesis [6,7]. The opponents of this view claim that in the chain of the pathological deformations leading to scoliosis, the ribs deform first and then the spine follows [8,9]. The rib cage deformity can be traditionally assessed radiographically using the rib vertebra angles (RVAs), mainly in early onset scoliosis, if it is developed before 10 years of age. A prognostic indicator for the progression of the curve is the difference of the concave minus the convex apical thoracic RVA (RVAD), if this is more than 20 degrees [10,11].

Regarding the frontal plane (FP), it was revealed by [12], that the scoliotic spine first deforms at the level of the intervertebral disc (IVD), not the vertebrae. These findings were confirmed three years later by [13]. Moreover, it was published that the histological abnormalities in the IVD in AIS are secondary to an altered mechanical environment [14,15,16,17,18]. These findings were based on the CTs of AIS patients, which showed that the IVDs contribute more to anterior spinal length compared to the contribution by the vertebrae, and suggested that the curve progression is not a primary vertebra growth disorder; rather, it is the result of altered mechanical loading.

The lateral spinal profile (LSP) and its significance in IS scoliogenesis is a topic that was discussed by many researches for many years, e.g., [19,20,21,22,23,24,25,26,27,28,29,30,31,32,33,34,35,36,37,38,39].

An answer to the question of whether emerging IS changes during growth are endogenous to or outside the vertebrae may be provided comparing the morphology of the growing spine during the early stages of IS development to normal peers’ spines.

It can be determined whether developmental changes in the spine are intrinsic or secondary by studying the various elements of the morphology of the growing spine during the early stages of IS development and correlating them with each other. This study may reveal the real role of the LSP in IS patho-biomechanics. Moreover, it may be helpful for tailoring an aetiological rather than symptomatic treatment, which is currently the case. This could be a medical treatment leaving the spine mobile and unfused.

Thus, the aim of this study is to address the above question by studying the sagittal profile of the onset and mild IS, using the radiography, the surface topography and the scoliometer readings of children with IS, examined in the scoliosis clinic of our department. We compared findings of these different examination methods and assessed their agreement or disagreement.

## 2. Materials and Methods

We performed a retrospective study of children and adolescents (16 years and younger), referred to the scoliosis clinic of our hospital by our school scoliosis screening (SSS) program, between January 2013 and December 2019. During this period, 2512 children attending schools located in the district of our hospital were screened.

In order to proceed with the study, we obtained approval by the hospital’s ethical committee (excerpt from the 24th on 6 October 2021 Ethical committee meeting) to conduct a retrospective study of radio-morphology of the spine, the scoliometer readings and the surface topography of patients attending our hospital scoliosis clinic.

Informed consent was also obtained by the parents of children for the examination.

### 2.1. The Examined Children

We studied two groups which included 43 children, 13 (30.2%) boys and 30 (69.8%) girls, with an average age of 12.7 ± 1.9 years (range 8.4–16.4 years).

The group A (IS group) included 17 children with IS, 15 girls and 2 boys. All of them had a trunk asymmetry, measured with a scoliometer, greater than or equal to 5 degrees, and Cobb angle more than 10 degrees in postero-anterior radiographs.

The group B (control group) included 26 children, 15 girls and 11 boys, with no trunk asymmetry, scoliometer measurement equal or less than 2 degrees, and straight spines in 4D Formetric.

### 2.2. The Measurements

The height and weight of children were measured. The Prujis scoliometer was used to examine the children in standing forward bending position (Adam test) in thoracic (T), thoraco-lumbar (TL) and lumbar (L) regions. The Cobb angle in the scoliotic children was assessed in the postero-anterior radiographs. The posterior truncal surface topogram, using the “Diers-Formetric 4” apparatus, was also performed.

The lateral truncal projection produced with the Diers 4D Formetric topogram is demonstrated in Figure 1. The lateral projection corresponds to the midline of the spine in the lateral view. This curve forms the 3D reconstruction of the spinal midline on conjunction with the frontal projection. The continuous green curve represents the external back profile (i.e., the symmetry line in the lateral view). The symmetry lines and anatomical landmarks are displayed in addition to the profiles. The symmetry line divides each profile into two halves with minimal lateral asymmetry. It approximates the line of the spinal process. In a healthy individual, the symmetry line is a straight line through VP (Vertebra Promines) and DM (Dimple Middle). The kyphotic apex is marked KA and the lordotic apex LA as seen in the diagram in lateral projection in Figure 1 and Figure 2.

Using the ruler on the left of the projection of the lateral profile, the distances VP up to KA and VP up to LA are measured in mm. Then a quotient is formed with numerator VP-KA and denominator VP-LA.

The distance from the vertebra prominence (VP) to the apex of the kyphosis (KA) and similarly to the apex of the lumbar lordosis (LA) was calculated on the topograms. The quotient of the distances (VP to KA) divided by (PV to LA) was also calculated.

The distance of the apparatus from the examined child was set at 2 m, the temperature of the examined room is the standard indoors temperature of 23–24 degrees Celsius, with a normal humidity of 36–38% so that the child was comfortable. To reduce error caused by potential movement of the child during the examination, the apparatus was designed to shut eight pictures in 10 s and the final result was the mean of all these captures.

### 2.3. Statistical Analysis

Power analysis was performed in order to assess significant differences in the variable of interest (Kyphotic Apex KA/Lordotic Apex LA) higher than 0.1 with a standard deviation of 0.1.

The averages of the parameters were studied, and the correlation of the ratio of distances (VP-KA) to (VP-LA) with the scoliometer and Cobb angle measurements was assessed respectively (Pearson corr. Coeff. r).

## 3. Results

Power analysis showed that in order to assess significant differences in the variable of interest (Kyphotic Apex KA/Lordotic Apex LA) higher than 0.1 with a standard deviation of 0.1, a sample size of at least 17 individuals in each study group (control and experimental) would be needed in order to achieve statistical power of 80% at a significance level (alpha) of 0.05. Descriptive statistics by group of interest and by gender are shown in Table 1.

### 3.1. Group A (IS)

The mean age of the girls was 12.4 (range: 8.4–14.3 years) with seven Major T curves (five right and two left) and eight Major TL, eight of which were double curves. The average height was 1.54 m (range: 1.37–1.64) and the average weight was 46.6 kg (range: 33–65). The mean T Cobb angle measurement was 29.28 degrees (range: 23–38 degrees) and TL 31.25 degrees (range: 21–37 degrees).

The mean scoliometer measurement was 7.47 ± 3.76 degrees.

The mean age of the boys was 15.3 (range:14.1–16.4 years) with a right Major T curve with a TL and a left TL curve. The average height was 1.69 m (range:1.66–1.71) and the average weight was 56.5 kg (range: 56–57). The Cobb angle measurements were T 36 degrees and TL 20 degrees. The mean scoliometer measurement was 12.5 ± 3.54 degrees.

Regarding the whole group, the mean (VP-KA) distance was 125.82 mm (range: 26–184) and the mean (VP-LA) distance was 321.7 mm (range: 237–417).

### 3.2. Group B (Controls)

The mean age of the girls was 12.2 (range: 9.3–15.2 years). The average height was 1.56 m (range: 1.39–1.73) and the average weight was 45.6 kg (range: 33–59).

The mean age of the boys was 13 (range: 9.9–15.6 years). The average height was 1.65 m (range: 1.38–1.84) and the average weight was 55.6 kg (range: 29–105).

Regarding the whole group, the mean (VP-KA) distance was 144.77 mm (range: 77–226) and the mean (VP-LA) distance was 329.038 mm (range: 282–426).

The ratio did not differ significantly between the two groups in our sample (0.39 ± 0.11 vs. 0.44 ± 0.08, *p*: 0.134). In the control group, KA (VPDM))/(LA (VPDM) was not significantly correlated with scoliometer findings (Pearson r = −0.016, −*p*: 0.939). The correlations of the ratio of distances (KA (VPDM))/(LA (VPDM)) with the measurement and scoliometer findings were non-statistically significant (Pearson r = 0.077, −0.211, *p*: 0.768, 0.416, respectively).

## 4. Discussion

As regards the radiological examination of the asymmetric children referred to our scoliosis clinic, it is noted that according to our recent practice policy, we do not prescribe radiological examination in younger asymmetric referrals (hump less than 4–5 degrees and children younger than 12–13 years of age). In these cases, we usually base our assessment on the surface topography picture. Radiological examination is only prescribed if the child’s asymmetry increases in the next visit to the scoliosis clinic [35].

It has been suggested that in IS, hypokyphosis is aetiological to the development of the deformity. The apex of the hypokyphosis is located in a vertebra located in a more cephalad level. In other words, the quotient numerator (VP-KA) decreases, and the quotient has a lower value. In that case, if the hypokyphosis were aetiologically responsible for the development of scoliosis, then the value of the quotient would have a negative statistically significant correlation with the Cobb angle and with the thoracic hump, too.

The results of the present report indicate that there is a negative correlation of the value of the ratio (VP-KA)/(VP-KA) with the scoliometer and Cobb angle measurements; yet it is not non-statistically significant (Pearson −0.356238) with a poor correlation coefficient. Similarly, the correlation with the scoliometer measurements is non-statistically significant (Pearson r = −0.2261). This means that both the transverse plane deformity of the thorax as expressed by the scoliometer measurements and the sagittal profile of the spine are not correlated with the Cobb angle, in mild and moderate scoliosis.

Additionally, this ratio did not differ significantly between the two groups in our sample (0.39 ± 0.11 vs. 0.44 ± 0.08, *p*: 0.134).

The opinion of the authors is that in initiating and mild scoliosis, the patho-biomechanics are probably dissimilar from the biomechanics when the curve is severe. This issue was studied in an earlier research project of our group in scoliotic children, and provided a clue to the question whether there is an intrinsic vertebral body growth disorder or not in mild/moderate IS. It was found that the sagittal profile of these IS curves do not differ significantly from the profile of normal peers, see: [36]. In other words, the growth potential in mild/moderate IS in the sagittal plane (the lateral spinal profile), is similar to that of peers having normal spines, in both vertebral bodies and the intervertebral discs (IVDs).

Regarding the role of hypokyphosis and its importance in IS aetiology, we stated earlier that in this report the minor hypokyphosis of the thoracic spine and its minimal differences observed in the small curves studied compared with that of non-scoliotics. This adds to the view that the reduced kyphosis, by facilitating axial rotation, could be viewed as being permissive, rather than as aetiological, in the pathogenesis of idiopathic scoliosis [36]. In other words, a straight (not bended) beam is more easily rotated than a curved one!

It may be stated that the study could be framed as a new scientific research of high clinical interest. Nevertheless, as a limitation it might be stated that for many years researches have been carried out in this sense and even if specific research features are present they are not the exclusive prerogative of this study which among other things used an evaluation parameter, (the surface topography), that is always carefully managed in scientific works, as although not invasive (total absence of radiation) in the application it often results to have variables that can subvert the study, as among other things admitted in the comparisons with other methods by the same authors.

## 5. Conclusions

This study indicates that hypokyphosis is not a primary causal factor for the commencing, mild or moderate scoliotic curve, as published elsewhere. We consider that the small thoracic hypokyphosis in developing scoliosis is considered as a permissive factor, rather than a causal one, in the pathogenesis of IS. This view is consistent with views previously published [5].

## Figures and Tables

**Figure 1 jcm-10-05901-f001:**
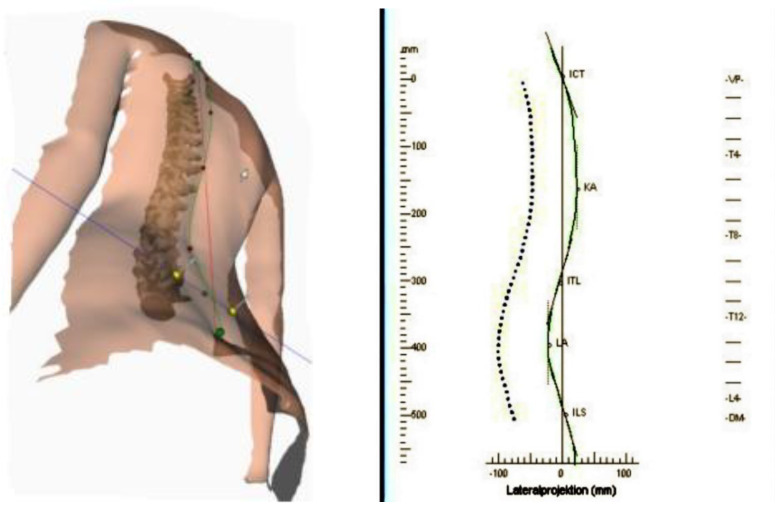
ICT or VP = vertebra prominence, KA = Kyphotic Apex, LA = Lordotic Apex, DM = Dimple Middle.

**Figure 2 jcm-10-05901-f002:**
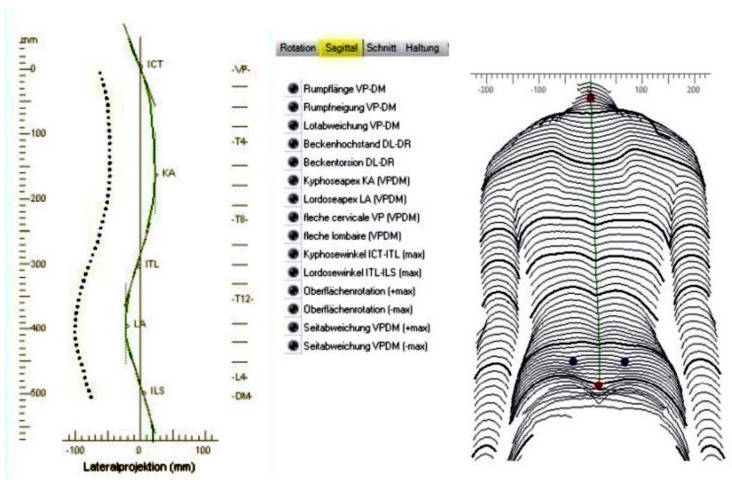
VP = Vertebra prominence, this corresponds to ICT (zero 0 mm), KA = Kyphotic Apex, LA = Lordotic Apex. If the (VP-KA) distance is shorter, that is, if the IS thoracic vertebra corresponding to KA is higher at the thoracic spine, then the value of the distances’ ratio (VP-KA)/(VP-LA) turns smaller, because the nominator of the quotient is smaller, (shorter VP-KA) distance. If this ratio were statistically significantly correlated with the Cobb angle and the truncal asymmetry, this would imply that this morphology (higher KA in IS) would be a primary aetiological factor for the commencement of IS. But this is not the case!

**Table 1 jcm-10-05901-t001:** Descriptive statistics by group.

	Group							
	Controls				IS			
	Gender				Gender			
	Boys		Girls		Boys		Girls	
	Mean	SD	Mean	SD	Mean	SD	Mean	SD
Age (years)	13.03	2.09	12.16	1.72	15.25	1.65	12.43	1.63
Height (cm)	165.14	16.36	155.55	10.28	168.50	3.54	154.00	8.47
Weight (kg)	55.64	22.43	45.55	8.86	56.50	0.71	46.60	8.93
Major Curve Cobb angle					28.00	11.31	30.33	9.22
Scoliometer	0.73	1.01	0.27	0.70	12.50	3.54	7.47	3.76
Kyphotic apex KA(VPDM)	−150.00	32.21	−140.93	35.90	−118.00	26.87	−126.87	36.48
Lordotic apex LA (VPDM)	−350.82	43.91	−313.07	48.61	−395.50	30.41	−311.80	37.20
Kyphotic Apex KA/Lordotic Apex LA	0.43	0.07	0.45	0.09	0.30	0.09	0.41	0.11

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
