# Peer review of "Morphology, Development and Deformation of the Spine in Mild and Moderate Scoliosis: Are Changes in the Spine Primary or Secondary?"

_jcm, 2021, doi:10.3390/jcm10245901_

Round 1

Reviewer 1 Report

In the presented research article, the authors try to answer the question, whether changes in the spine morphology are primary or secondary. School children’s screenings were analyzed.

The authors conclude that hypokyphosis is not a causal factor for a scoliotic curve. Although negative results – which disprove the hypothesis are generally more difficult to publish – this is worthy of mention!

Too much is still unclear about the development of idiopathic scoliosis and the necessity of its treatment.

I like the paper. However, minor revision of the introduction and methods is necessary:

  • Please, no citations in the abstract

Introduction:

  • In general, this paragraph needs more structure, I suggest:
  • ) what is idiopathic scoliosis – incidence?
  • ) changes in the frontal plane – how can they be measured?
    • Cobb Angle
    • Please mention RVAD as well – see Publication by Metha or Ballhause TM, BMC Musculoskeletal Disorders. 2019

3.) Changes in the sagittal plane – the lateral spinal profile – less citations in this paragraph – Can a lateral spinal profile be defined in children?

4.) Hypothesis of the article

  • Line 64: “on this” is double

Methods:

  • How many children in total were screened?
  • In the control group B are 15 girls and 11 boys, I recommend the same gender distribution as in group A.
  • Were x-rays performed to confirm pathologic results?
  • The explanation of the Diers 4D Formetric topogram (ll. 189-198) should appear in the methods and not the discussion of the paper

Author Response

Dear Editor in Chief

Thank you very much for inviting us to submit a revised version of our manuscript: (JCM -1497076) entitled: " Morphology, development & deformation of the spine in mild and moderate scoliosis: Are the changes in spine primary or secondary?".

We express our appreciation to the reviewers for taking their time and expertise to review our submission and suggest their comments to improve our submission.

We have checked our manuscript according to the reviewers’ recommendations. We have responded to their comments point by point in red color letters, and we submit a revised manuscript file.

We would be very grateful if you could consider our manuscript to be published in your journal.

Response to Reviewer 1 Comments

In the presented research article, the authors try to answer the question, whether changes in the spine morphology are primary or secondary. School children’s screenings were analyzed.

The authors conclude that hypokyphosis is not a causal factor for a scoliotic curve. Although negative results – which disprove the hypothesis are generally more difficult to publish – this is worthy of mention!

Response: We thank the reviewer for this statement.

Too much is still unclear about the development of idiopathic scoliosis and the necessity of its treatment. I like the paper. We thank the reviewer for these kind words. However, minor revision of the introduction and methods is necessary:

  • Point 1. Please, no citations in the abstract

Response: We thank the reviewer for this statement. The citations were deleted.

Introduction:

  • In general, this paragraph needs more structure, I suggest:
  • what is idiopathic scoliosis – incidence?
  • Response: Thank you. We added a phrase on incidence.
    • changes in the frontal plane – how can they be measured? Cobb Angle

Please mention RVAD as well – see Publication by Metha or Ballhause TM, BMC Musculoskeletal Disorders. 2019

Changes in the sagittal plane – the lateral spinal profile – less citations in this paragraph – Can a lateral spinal profile be defined in children?

Response:  Thank you. We added 1) a paragraph describing the way of assessment of the changes in the frontal plane using the Cobb angle and the related literature, 2) information about the lateral profile assessment and the rib cage deformity which may be assessed using the RVAs. The value of RVADs are also mentioned and the related literature is added.

4.) Hypothesis of the article

  • Line 64: “on this” is double

Response:  Thank you. The double “on this” was deleted.

Methods:

  • How many children in total were screened?
  • Response: Thank you. We note the number of screened children.
  • In the control group B are 15 girls and 11 boys, I recommend the same gender distribution as in group A.
  • Response: Thank you: We respect your suggestion as there is a different distribution in gender between the two groups of interest. We would like to notice that our study evaluates a relatively novel subject and we would like to include as many participants as possible in order to increase the statistical power. However, in order to improve the readability of the paper, table 1 is now more concise, describing the sample by gender and group of interest. The results follow the same discrimination in order to clarify the conclusions.
  •  
  •  
  • Were x-rays performed to confirm pathologic results?

Response: Thank you. In group A the deformity was assessed prescribing radiographs. In Group B the children had minimal or no asymmetry (hump) therefore we do not prescribe radiographic examination.

We note that during the period of recent years SSS, in younger asymmetric referrals (hump less than 5-4 degrees and children younger than 13-12 years of age), it is not our policy to prescribe radiological examination,  but only if the asymmetry is increased in the next visit to our scoliosis clinic, see our publication: The effect of growth on the correlation between the spinal and rib cage deformity: implications on idiopathic scoliosis pathogenesis. Grivas TB, Vasiliadis ES, Mihas C, Savvidou O.Scoliosis. 2007 Sep 14;2:11. doi: 10.1186/1748-7161-2-11. In these cases, we usually base our assessment on the surface topography examination. This statement is added in the discussion section.added in the discussion section.

  • The explanation of the Diers 4D Formetric topogram (ll. 189-198) should appear in the methods and not the discussion of the paper

Response: Thank you. The explanation of the Diers 4D Formetric topogram (ll. 189-198) now appears in the methods section.Τέλος φÏŒρμας

Response to Reviewer 2 Comments

Point 1. The study could be framed as a new scientific research of high clinical interest, but nevertheless for many years researches have been carried out in this sense and even if specific research features are present they are not the exclusive prerogative of this study which among other things used a evaluation parameter that is always carefully managed in scientific works, as although not invasive (total absence of radiation) in the application it often results to have variables that can subvert the study, as among other things admitted in the comparisons with other methods by the same authors.

Response: Thank you for the statement. We have included this opinion in our discussion as potential limitation of the study. See in the text. “It may be stated that the study could be framed as a new scientific research of high clinical interest. Nevertheless, as a limitation it might be stated that for many years researches have been carried out in this sense and even if specific research features are present they are not the exclusive prerogative of this study which among other things used an evaluation parameter, (the surface topography), that is always carefully managed in scientific works, as although not invasive (total absence of radiation) in the application it often results to have variables that can subvert the study, as among other things admitted in the comparisons with other methods by the same authors.”

Point 2. Good materials and methods (although in my opinion the population to be evaluated should be significantly increased), good statistical study, clear exposure but the study nevertheless confirms what has been highlighted by other studies since 2014. Response: Thank you for kind words and for the statement noted.

Point 3. It would also be useful for the authors to define in practice the acquisition method with the Formetric (distances, temperatures, humidity, etc. all factors that modify the results)

Response: Thank you. We define in practice the acquisition method with the Formetric in the methods section. See text. “The distance of the apparatus from the examined child is set at 2 meters, the temperature of the examined room is the standard indoors temperature of 23-24 degrees Celsius, with a normal humidity of 36-38% so that the child is comfortable. To reduce error caused by potential movement of the child during the examination, the apparatus is designed to shut eight pictures in 10 seconds and the final result is the mean of all these captures.

Finally, we note that the paper was reviewed linguistically by a native English speaker.

Reviewer 2 Report

the study could be framed as a new scientific research of high clinical interest, but nevertheless for many years researches have been carried out in this sense and even if specific research features are present they are not the exclusive prerogative of this study which among other things used a evaluation parameter that is always carefully managed in scientific works, as although not invasive (total absence of radiation) in the application it often results to have variables that can subvert the study, as among other things admitted in the comparisons with other methods by the same authors. good materials and methods (although in my opinion the population to be evaluated should be significantly increased), good statistical study, clear exposure but the study nevertheless confirms what has been highlighted by other studies since 2014. it would also be useful for the authors to define in practice the acquisition method with the Formetric (distances, temperatures, humidity, etc. all factors that modify the results)

Author Response

(The authors gave the same response as above.)
